# Antioxidation Defenses of *Apis mellifera* Queens and Workers Respond to Imidacloprid in Different Age-Dependent Ways: Old Queens Are Resistant, Foragers Are Not

**DOI:** 10.3390/ani11051246

**Published:** 2021-04-26

**Authors:** Jerzy Paleolog, Jerzy Wilde, Artur Miszczak, Marek Gancarz, Aneta Strachecka

**Affiliations:** 1Department of Zoology and Animal Ecology, Faculty of Environmental Biology, University of Life Sciences in Lublin, Akademicka 13, 20-950 Lublin, Poland; aneta.strachecka@up.lublin.pl; 2Department of Poultry Science and Apiculture, Faculty of Animal Bioengineering, Warmia and Mazury University in Olsztyn, ul. Słoneczna 48, 10-957 Olsztyn, Poland; jerzy.wilde@uwm.edu.pl; 3Food Safety Laboratory, The National Institute of Horticultural Research, Pomologiczna 13b, 96-100 Skierniewice, Poland; artur.miszczak@inhort.pl; 4Institute of Agrophysics, Polish Academy of Sciences, Doświadczalna 4, 20-290 Lublin, Poland; m.gancarz@ipan.lublin.pl

**Keywords:** honey bees, *Apis mellifera*, oxidative stress, imidacloprid, social evolution, aging

## Abstract

**Simple Summary:**

Honey bees are unique for studies on aging because queens live 40-fold longer than workers. An efficient antioxidant defense (ADS) is thought to be pivotal for longevity, but not always. How were different ADSs shaped by evolution in young and old queens and workers? Honey bees, the essential pollinators, are facing depopulation due, at least in part, to pesticides, such as imidacloprid, an oxidative stressor. Is an evolutionarily shaped ADS still useful for contemporary young and old queens/workers? Answering these questions is important for emerging oxidative-stress ecology and protecting contemporary honey bees. The ADS activity was determined in 1-day-old, 20-day-old, and 2-year-old queens and in 1-day-old and 20-day-old workers (foragers) fed without (control) or with low or high imidacloprid (in bee food). ADS was upregulated in workers with age but downregulated in queens. However, imidacloprid oxidative stress suppressed the active ADS in workers, particularly 20-day-old foragers, but not in 1-day-old queens. Unexpectedly, poor ADS activity in 2-year-old queens was highly upregulated by imidacloprid. Thus, queen and worker ADSs respond to imidacloprid in opposite ways, and old queens were still resistant, but foragers were not. This may be unfavorable for foragers dwelling in ecosystems that expose them to pesticides.

**Abstract:**

We investigated how different antioxidant defenses (ADSs) were shaped by evolution in young/old *Apis mellifera* workers and queens to broaden the limited knowledge on whether ADSs are effective in contemporary pesticide environments and to complete bee oxidative-aging theory. We acquired 1-day-old, 20-day-old, and 2-year-old queens and 1-day-old and 20-day-old workers (foragers) fed 0, 5, or 200 ppb imidacloprid, a pesticide oxidative stressor. The activities of catalase, glutathione peroxidase, glutathione S-transferase, and superoxide dismutase and the level of total antioxidant potential were determined in hemolymph. The ADS was upregulated in workers with age but downregulated in queens. Imidacloprid suppressed the ADS in all workers, particularly in foragers with an upregulated ADS, but it did not affect the ADS in 1-day-old queens. In contrast to foragers, the downregulated ADS of 2-year-old queens was unexpectedly highly upregulated by imidacloprid, which has not been previously shown in such old queens. The principal component analysis confirmed that queen and worker ADSs responded to imidacloprid in opposite ways, and ADS of 2-year-queens was markedly different from those of others. Thus, evolutionary shaped ADSs of older queens and workers may be of the limited use for foragers dwelling in pesticide ecosystems, but not for old queens.

## 1. Introduction

The appearance of oxygen in the Earth’s atmosphere was ecologically and evolutionarily relevant [1]. Aerobic organisms emerged, and natural selection pressed them to develop antioxidant defense systems (ADSs) to minimize the destructive impact of oxidative metabolism. Oxidative stress is also a significant modulator of life-history strategies [2], including in honey bees (*Apis mellifera*). However, there is limited evidence that this modulation could change after thousands of years of natural selection, as honey bees have faced anthropogenically contaminated environments.

There are indications of a link between the individual life-span and the efficiency of ADSs [3]. This has resulted in the free radical theory of aging, which states that harmful effects of reactive oxygen species (ROS) accelerate senescence [4,5]. On the other hand, ROS have been shown to extend the life-span by acting as pro-survival signals [6,7]. In this context, keeping in mind that contemporary honey bees are exposed to anthropogenic, xenobiotic stressors, the suggestion that ROS accelerate senescence predominantly under stress conditions [8] seems to be important. Honey bee queens and workers have different life-history strategies, evolving very different life-span phenotypes; therefore, they are considered to be an excellent model for studying different aspects of aging [9,10]. We hypothesize that natural selection had to shape not only different ADSs, but ADSs of highly differing and even opposite activities in queen and worker bees, particularly when considering old workers vs. old queens (H1). Moreover, if eusocial evolution has shaped queens and workers for different longevity and ADS phenotypes, it is necessary to better understand how these different ADS phenotypes fit the anthropogenically contaminated (e.g., pesticides), contemporary environment, if they fit in at all (H2). We assume that it is not the baseline, the evolutionary activity of ADSs that is important for honey bee fitness but how strongly they can be activated by the xenobiotic, oxidative stress nowadays. Consequently, the aim of this research was to study the differences between the efficiency of the ADS in young vs. old workers and queens when unexposed (control; aging only; H1) or exposed to low and high pesticide oxidative stress (effect of caste × aging × stress; evolution story vs. contemporary stress; H2). The ADSs of queens and workers were evaluated when aging [2,5], and the ADSs of workers when exposed to pesticides [3,8]. However, the complex comparative experimental design proposed here will help fill gaps in understanding the variation in oxidative status and oxidative stress, including the adaptive response of the ADS to pesticides, which is important to answer questions on emerging oxidative stress ecology [7]. We also expect that results obtained this way may help protect contemporary depopulating honey bees. To the best of our knowledge, we are the first to test 2-year-old queens kept in commercial colonies.

Imidacloprid (ID) [1-(6-chloro-3-pyridylmethyl)-2nitroimino-imidazolidine], which honey bees are exposed to by agriculture, has been found to be harmful to honey bees’ field navigation and health [11,12,13], including their detoxification abilities and reproductive functions [14,15,16,17]. Moreover, ID impairs the honey bee ADS and suppresses antioxidant genes [3], simultaneously increasing MDA (a marker of oxidative stress; lipid peroxidation) and shortening workers’ life-spans [18,19]. Consequently, ID reduces the colony’s fitness. The globally used ID, which stimulates numerous discussions and researches, belongs to neonicotinoid insecticides that are implicated as one of the major reasons for declines in global bee populations. Honeybees (essential pollinators worldwide) are susceptible to even potentially sublethal exposure to neonicotinoids [12]. Therefore, we used ID as a common contemporary oxidative stressor.

## 2. Materials and Methods

This study was performed in 2018 and 2020 in the Warmia and Mazury region of Poland (19.53 E, 53.50 N) under field conditions and at a location where natural bee food was almost absent within the range of forager flights during the research.

### 2.1. Rearing of Workers

Three groups of five *Apis mellifera carnica* colonies (2 supers; frame 360 mm × 260 mm) populated by similar numbers of bees and with similar structures were given sugar-water syrup (5:3 w/w) with 0 ppb (I-0; control), 5 ppb (I-5), or 200 ppb (I-200) ID (Bayer Health Care AG, Leverkusen, Germany). The bees were also fed a mixture of pollen and commercial bee food (API-Fortune HF 1575, Bollène, France), which, like the syrup, contained 0 ppb, 5 ppb, or 200 ppb ID; 5 ppb is close to field-relevant sublethal residual concentration, whereas 200 ppb is considered potentially lethal [11,17]. The queens of all of the colonies originated from the same commercial stock.

After 1 month of such feeding, each queen was placed in a 2-comb cage made of queen-excluder for 24 h and then released. The combs containing the newly laid eggs were left in their colonies within the cages for further development. Before worker emergence, these combs were transferred from the hives to an incubator (34.5 °C), where the emerging workers were captured at 6 h intervals. Thus, the 1-day-old workers were either immediately used for biochemical testing or, after marking them with the colony-specific colors, transferred to their original colonies. When they reached the age of 20 days, they were recaptured and used for biochemical testing.

### 2.2. Rearing of Queens

After completing the worker assays, three queen-less nuclei (queen banks), each populated by workers from their own group, were created in I-0, I-5, and I-200 and fed the same group diet. One-day-old larvae were grafted from the original I-0, I-5, and I-200 colonies into the nuclei belonging to the same group. Queen cells obtained in this way were individually incubated (34.5 °C) to obtain 1-day-old virgin queens. These queens were treated in one of three ways: immediately used for biochemical testing; returned to their source nuclei from which they were taken for the biochemical testing when they reached the age of 20 days (still virgin); or introduced into commercial colonies, where they remained for the following 2 years. Two-year-old queens obtained this way were then removed from these colonies and introduced to other queen-less colonies kept in fully populated mini-plus hives (6 frames of 251 mm × 159 mm in one super). Three groups, each of 13 mini-plus colonies, were established in this way and subsequently fed the I-0, I-5, or I-200 diet for the following 4 weeks. After that, the queens were captured for biochemical testing.

### 2.3. Imidacloprid Contamination

For 1-day-old and 20-day-old queen/worker hives, the concentration of ID in the syrup (3 months after preparation) amounted to 4.2 ppb and 196 ppb in I-5 and I-200, respectively. This amounted to (mean ± SD) 0.35 ± 0.24 ng/bee in the bodies of approximately 100 workers at 1–10 days of age from each rearing colony in I-200.

For 2-year-old-queen hives, the concentration of ID in the syrup from the hive feeders amounted to a mean ± SD of 0.0 ± 0.0 ppb in I-0, 4.9 ± 0.32 ppb in I-5, and 186.2 ± 0.32 ppb in I-200 (*n* = 13 hive samples per diet). The concentration of ID in the comb storage was 0.0 ± 0.0 ppb in I-0, 4.1 ± 0.51 ppb in I-5, and 111.7 ± 56.33 ppb in I-200 (*n* = 13 hive samples per diet). The ID concentration in the bodies of 10-day-old workers was 0.0 ng/bee in I-0 and I-5 and 0.48 ± 0.38 ng/bee in I-200 (*n* = about 100 bees from each colony).

Liquid chromatography-tandem mass spectrometry was applied (see Słowińska et al. [19]).

### 2.4. Laboratory Procedures

The hemolymph was individually sampled from 10 workers and 5 queens within each colony/nuclei within each group (I-0, I-5, and I-200) and two age classes (1-day-old and 20-day-old) using 20 µl glass capillaries (“end-to-end” type; without anticoagulant; Medlab Products; Raszyn, Poland) inserted between the third and fourth tergite. Capillaries with the hemolymph of workers and queens belonging to a particular colony were placed separately into two sterile Eppendorf tubes containing 150 µl of ice-cooled 0.6% NaCl in order to obtain the pooled queen and pooled worker samples. The tubes were immediately frozen at −80 °C for further biochemical analyses. This resulted in 30 pooled samples of queen hemolymph (2 age groups × 15 colonies) and 30 pooled samples of worker hemolymph (2 age groups × 15 colonies). In the 2-year-old queens, hemolymph was taken individually from 13 queens within each of the I-0, I-5, and I-200 groups (3 groups × 13 = 39 samples) and frozen at −80 °C. The detailed protocols were described by Łoś and Strachecka [20].

All antioxidant enzyme activities were calculated per 1 mg of protein as follows: catalase (CAT) by the method described by Aebi [21], glutathione peroxidase (GPx) by the method in Chance and Maehly [22], glutathione S-transferase (GST) by the method n Warholm et al. [23], and superoxide dismutase (SOD) by the method in Podczasy and Wey [24]. The total antioxidant potential measured by determining the ferric reducing antioxidant potential (FRAP) was determined using the Benzie and Strain [25] method. All methods used for determining the activities of CAT, GPx, GST, SOD, and FRAP were applied by taking into account the modifications developed by Strachecka et al. [26] and Łoś and Strachecka [20]. CAT, GPx, GST, and SOD are the major ROS scavenging and antioxidant enzymes in honey bees [18,19], and they constitute the first line of their ADS barrier.

### 2.5. Statistical Analysis

ANOVA accompanied by the LSD test (*p* ≤ 0.01) was applied in order to estimate and compare the effects of bee caste (queen, worker), bee age (1-day-old, 20-day-old, and 2-year-old) (H1), and ID dose (I-0, I-5, and I-200) (H2); the ID dose was nested within age and caste. Principal component analysis (PCA), including correlations matrix, was performed by taking into account the effects of bee caste and age (H1) vs. ID dose (H2) on the main variation components (PC1 and PC2), which were determined based on the Cattell criterion. All statistical analyses were performed in Statistica version 12.0 (StatSoft Inc., Tulsa, OK, USA) with *α* = 0.05.

## 3. Results

The activities of ADS enzymes and FRAP increased with age in I-0 workers (Figure 1 and Figure 2). In contrast, in queens, the activities of ADS enzymes decreased with age, whereas FRAP levels were similar in the 1-day-old and 20-day-old queens but markedly increased in 2-year-olds.

When bees were exposed to ID, SOD was downregulated in workers and queens at all ages, though less in I-5 and more in I-200 (Figure 1). The remaining ADS enzymes and FRAP (Figure 2) were downregulated by ID in 20-day-old workers in proportion to the dose, but in 1-day-old workers, this occurred only in the I-200 group. In the 1-day-old workers from I-5, ID upregulated the activities of CAT, GSP, and GPx and downregulated FRAP.

In contrast, ID did not downregulate but rather upregulated FRAP and ADS enzymes in 20-day-old and 2-year-old queens. The enzymes were upregulated but FRAP downregulated by ID in 1-day-old queens. Therefore, the FRAP response was opposite the enzyme response in 1-day-old queens (Figure 2) but similar in the older queens. Importantly, the ADS of older queens and workers, in particular, responded to ID stress in a different, frequently even opposite, manner; worker ADSs were suppressed, but queen ADSs activated.

Two main PCA components defined 80.16% of the variation (PC1, 40.07% and PC2, 33.12%). CAT and GST were strongly positively correlated, and their correlation with GPx was weaker (Figure 3A). Those three enzymes did not correlate with FRAP and SOD, which highly negatively correlated with each other. CAT and GST markedly influenced PC1, whereas FRAP markedly influenced PC2. GPx strongly affected both PC1 and PC2, whereas SOD weakly influenced PC2. Positive values of PC2 described queens, whereas the negative values described workers (Figure 3B); the ADSs of workers and queens responded to ID stress differently, regardless of the influence of age and ID dose. Only one quadrant (negative PC1 and positive PC2) defined the 2-year-old queens (Figure 3C), whereas the other three quadrants defined the remaining age classes in workers and queens, confirming the otherness of the ADS of 2-year-old queens. The ID dose influenced neither PC1 nor PC2; no plots are presented. Taken together, PCA showed that caste had the greatest effect on ADS variability, age had less of an effect and was more involved in interactions, and the ID dose mattered only for the caste and age interactions. ANOVA confirmed that the caste effects were the greatest (SOD, CAT, GST, GPx, FRAP; F = 59.50, 46.88, 12.55, 21.92, 41.61; df = 1; *p* = 0.0000, 0.0000, 0.0008, 0.0000, 0.0000). The effects of age were significant but involved in age × caste interactions. Effects of ID dose were not so clear and often insignificant, as they were involved in caste × ID dose and age × ID dose interactions. This is in agreement with the PCA results.

## 4. Discussion

The following comments are important for the further discussion of our results: 1-day-old workers do not perform any special tasks yet, and 20-day-old workers end up their nest-bee tasks (young bees) and start their forager task (old bees); therefore, mainly the influence of age and not the social task is discussed. Both 1-day-old and 20-day-old queens were virgins; thus, their mating status did not impact the findings. Two-year-old queens cannot exist within a normal colony as virgins; therefore, the impact of age in their case has to be inextricably linked with their age status. The apian phenotypic plasticity allows workers to perform different behavioral tasks at the same age, but the biological senescence is slowed down, accelerated, or even reversed in such situations [5]. Therefore, the biological senescence is something different than ageing counted in days. Rather than ageing counted in days, our reasoning and conclusions concern the biological senescence.

Załuski et al. [27] observed the downregulation of SOD in bees exposed to a pesticide. The activities of SOD isoforms depend on the cell concentration of manganese, zinc, and copper ions [18]. Nikolić et al. [28] showed that pesticides disturb the composition of bioelements related to the activity and efficiency of honey bee SOD. This can cause the inactivation of SOD without affecting the activity of other antioxidant enzymes. SOD was the only ADS enzyme distinctly downregulated by ID in both our queens and workers of all ages. Therefore, the cytoplasmic and SOD-dependent mitochondrial conversion of O_2_ to H_2_O_2_ and O_2_ may be impaired by ID independent of bee caste and age, which is important for the fitness of the entire colony.

Supposing that eusocial evolution produced distinctly different life-span phenotypes for queens and workers [2,29] and that natural section forced honey bee queens and workers to evolve their ADSs [30], and if the free radical theory of aging is applicable to honey bees [31,32], one should expect distinct age-dependent differences between the ADS activities in workers and queens not only during their aging, but also differences when exposed to oxidative stress. Our study has confirmed this assumption, enriching the limited information on the consequences of bee-caste life-stories for dwelling bees in the contemporary, anthropogenically contaminated environment (H1 and, particularly, H2). Importantly, as longevity and reproductive abilities are evolutionarily linked, the concept of “life-span-phenotype” corresponds to the concept of “fecundity-phenotype”; this will be discussed further.

In workers, ID suppressed the ADS, particularly in the I-200 group, whereas it activated the ADS in queens, with the exception of SOD and FRAP in 1-day-olds. Assuming that increased activity of antioxidant enzymes leads to a corresponding increased resistance to oxidative stress [5,33], this study revealed that, while the queen organisms fought to overcome the ID stress by upregulating ADS enzymes, the worker organisms did it only partly in I-5 and not at all in I-200 (H2). Importantly, the 2-year-old queens unexpectedly increased the ADS activity to the greatest extent when exposed to ID. To the best of our knowledge, we analyzed the ADS of queens of such an advanced age for the first time, and their ADS still was very effective when exposed to ID despite the queen senescence. This finding is important for oxidative stress ecology [7] and better understanding bee caste senescence.

Both ADS enzymes and FRAP were upregulated with age in our workers (H1). Johnson and Carey [31] and Margotta et al. [32] suggested that, due to the oxidative stress caused by the foraging environment and working flight muscles, the most active ADSs should be in the foragers, but this was not the case in our 20-day-old workers exposed to ID, as their more active ADSs were seriously suppressed (H2). Due to impairment of the ADS by ID, longevity [3,11], resistance to parasitoses [13,34], and harmful xenobiotics [35] may also be decreased in the foragers. Thus, decreased forager ADSs due to decreasing colony survival may impact the colony’s fitness (compare Lemanski et al. [10]). Taken together, this explains how the worker (forager) ADS would be affected by natural selection, which could partially shift the direction of its pressure under the current severe oxidative pesticide stress. This issue requires further study.

The opposite, slightly surprising, phenomenon was found in queens; their ADS enzymes were downregulated and FRAP upregulated during the aging process (H1). Moreover, the queen ADS enzymes were not suppressed or even largely upregulated when exposed to ID. Again, these phenomena happened particularly in our 2-year-old queens (H2). Corona and Robinson [36] suggested that a decrease in antioxidant gene expression seems to be a general pattern in long-lived queens. On the other hand, Hsu and Hsieh [37] pointed out that the young queens had lower ROS levels, as well as lower levels of SOD, CAT, and GPx. However, it may be not so much the age-related changes in ADS activity that are crucial for contemporary honey bees as the ability of their ADS to cope with the severe oxidative stress. This is a new aspect of this discussion. Such abilities were possessed by our old queens but not by the foragers. Another important finding was that the downregulation of the ADS enzymes with aging in our queens was accompanied by FRAP upregulation. Although the activity of a specific antioxidative enzyme does not have to correlate with the total antioxidant capacity [38,39], which was also shown by PCA, it is not a sufficient explanation for different patterns of changes in ADS enzymes vs. FRAP during the ageing process in our queens. Honey bee ADSs consist of many protein and non-enzymatic elements and are related to the action of other compounds, including the proteolytic system, juvenile hormones, and lipids, among others [27], which have different expression levels in workers and queens [40]. Fat body antioxidative abilities and thioredoxin reductase [37], as well as methionine sulfoxide reductase, should be taken into consideration as well. The ratio of saturated to unsaturated fats in cell membranes promotes higher resistance of queens to oxidative stress [41,42]. Vitellogenin is also a strong antioxidant [9,43] with lower expression in workers, decreasing with age as opposed to queens, in which the expression increases as they progress in age [44,45]. Differences in the expression of ADS genes were also revealed during the individual development of these castes [46]. Therefore, the ADSs of our queens have to work differently and may involve many more different components than the ADSs of our workers during exposure to ID stress, particularly in the case of the oldest queens. This is in agreement with two evolutionary pathways proposed for understanding senescence: the free radical and adaptive senescence theories [31,42]. The response of the ADSs of our queens fits the free radical adaptation (compare Santos et al. [46]), whereas the workers fit adaptive senescence [31,32]. Consequently, eusocial evolution must have shaped queen and worker ABSs very differently (H1), but our study revealed that this could be not useful, and even unfavorable, for workers, particularly forgers that dwell in the anthropogenic ecosystems, especially when exposed to pesticides. On the other hand, 2-year-old queens seemed to be ready to deal well with pesticides.

To better support the above findings, we should consider the following: CAT, GPx, GST, and SOD are the major ROS scavenging, antioxidant enzymes in honey bees constituting the first line of the ADS barrier against xenobiotic oxidative stress factors. Their functions are limited to only this activity, and their secretion/activation is directly controlled by ROS. In addition to antioxidation, the other enzymes, e.g., vitellogenin, regulate other physiological processes and are adjusted by hormones, constituting the second ADS line [5,6,28,30,42]. The first line of defense in our foragers was suppressed by ID, even though its activity was increased by aging. On the contrary, the activity of the first line of defense in 2-year-old queens decreased with age. However, it was not suppressed but rather only “silenced,” as its activity retained the ability to be increased in response to ID. It is the age-related downregulation of the ADS enzymes in our queens, accompanied by FRAP upregulation, that is in line with this finding. This is very interesting, new evidence that requires further research. Consequently, it may be not so much the baseline activity of ADS that is important for longevity as how strongly the characteristics can be upregulated when exposed to oxidative stress.

The negative link between reproduction and longevity evolved in most solitary animals, as the organism (soma) becomes redundant when offspring has been produced, particularly while there is a shortage of resources that needs a number of trade-offs between body maintenance and reproductive functions. However, this link has been evolutionarily decoupled in social insects [2,31,47]; a positive correlation between reproduction and lifespan emerged; the fecund queens living several folds longer than even egg-laying workers [2,5]. The generation overlapping within a nest is thought to be the trigger factor here, as the females that live the longest (future queens) produce much more offspring than the short-living helper-females (future workers) [31]. However, to evolve, the long-living females, fully fecund throughout their entire life, should develop a positive linkage (in contrast to solitary organisms) between body maintenance and reproductive functions, and there is preliminary evidence that such a link exists [47,48]. It is the ADS that is the important element of the body maintenance abilities [7,9,32,41,49]. It becomes more and more important in the contemporary environments in which honey bees face colony depopulation [12,34]. Consequently, this study has provided new evidence for analyzing the co-evolution of longevity, fertility, social functions, and body maintenance abilities, including mutual trade-offs and genetic or functional inter-trait correlations. Our finding that the ADS was downregulated in most cases upon ID exposure in workers, but upregulated particularly in old queens, is pivotal here. It not only proves that the more reproductive, long-lived females (queens) have better body maintenance abilities (ADS) but also that they are ready to overcome the pesticide stress, even though they consume more protein-rich food (negative link of longevity-food intake). Therefore, positive evidence for an evolutionary link between high fecundity and longevity and the existence of mechanisms enabling the positive link between the antagonistic demands of reproduction and body maintenance [48] was provided. Our queens were able to upregulate their ADS despite their senescence and additional nutritional effort linked to reproduction [47], revealing the positive link between resistance to ID oxidative stress longevity and, maybe, nutritional effort. The contrary was found for our foragers which did not have such abilities, as they had a very different reproductive status and ADS barriers than our queens. Therefore, forager survival is more crucial for the colony’s fitness nowadays than it was before. On the other hand, the effect of ID in our young, 1-day-old workers was more queen-like than forager-like, which corresponds with the transcriptomic studies in *Bombus terrestris* [50]. This shows that reproductive abilities, no matter how real or expected, were crucial for the evolutionary shaping of the network of linkages: fecundity-longevity, body maintenance/defense, and particular phenotypes of these traits.

Our complex experimental protocol employing ageing of the honey bee castes vs. oxidative pesticide stress could be useful for studies of results of the biochemical evolution of the ADS vs. colony fitness [10], as the worker response to xenobiotics has predominantly been studied previously. This is also important for emerging oxidative stress ecology [7], as oxidative stress may play a role in the adaptive response to environmental pressure [51], including neonicotinoids [52]; therefore, it may also be important for better protection of contemporary honey bees. Notably, hemolymph was used in our study, and Sagona et al. [53] showed that the distribution of antioxidant enzymes differs depending on the tissue and organs of honey bees. Thus, our protocol appears to be a good tool for developing further knowledge on the ADSs of different organs and tissues.

## 5. Conclusions

The evolution of eusociality acting at the biochemical level has shaped queen and worker ADSs very differently, with the most serious consequences for older bees. The evolutionarily shaped ADS seems to be insufficient, or even unfavorable, for the forager workers, which currently dwell in anthropogenic ecosystems when exposed to pesticide oxidative stress. Despite their senescence, the ADSs of 2-year-old queens seem to perform unexpectedly well at overcoming this stress.

There is a positive, evolutionary link between fecundity-longevity and body maintenance/defense in the old queens but not in the forager workers.

ID is referred to as a “green” pesticide. However, it seems to be not so “green” due to its suppressive effects on forager ADSs and the age- and caste-independent impairment of SOD. Comparative studies on the response to ID or other xenobiotics, using the experimental protocol we proposed here, are needed to determine better honey bee protections (e.g., feed supplements) and to answer questions on emerging oxidative stress ecology.

## Figures and Tables

**Figure 1 animals-11-01246-f001:**
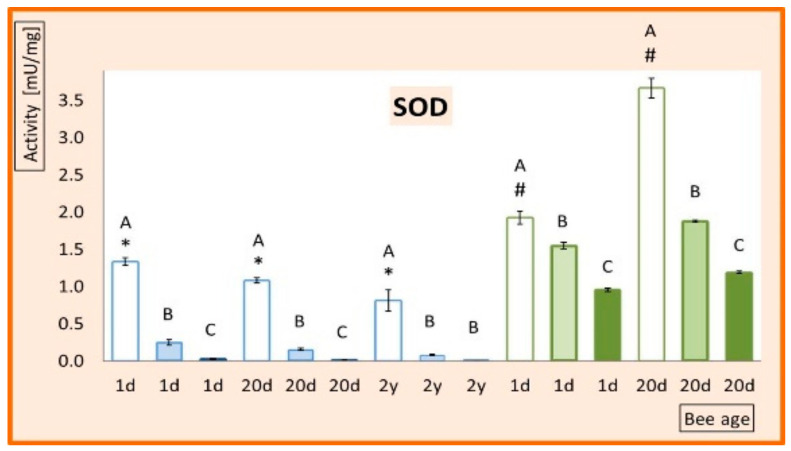
Mean activities of superoxide dismutase (SOD). The error bars indicate SD. Bee diet (group) contains 0 ppm (I-0), 5 ppm (I-5), and 200 ppm (I-200) imidacloprid. 1-day-old bees (1d), 20-day-old bees (20d), and 2-year-old bees (2y); queens (**blue bars**); workers (**green bars**); bee diet contains 0 ppm (**transparent bars**), 5 ppm (light-shadowed bars), and 200 ppm (dark-shadowed bars) of imidacloprid. Different capital letters indicate that differences between the means (diets) nested within a particular age and caste are significant (ANOVA + LSD; *p* < 0.01). Values marked with hash differ significantly from each other. Values marked with asterisk differ significantly from one another.

**Figure 2 animals-11-01246-f002:**
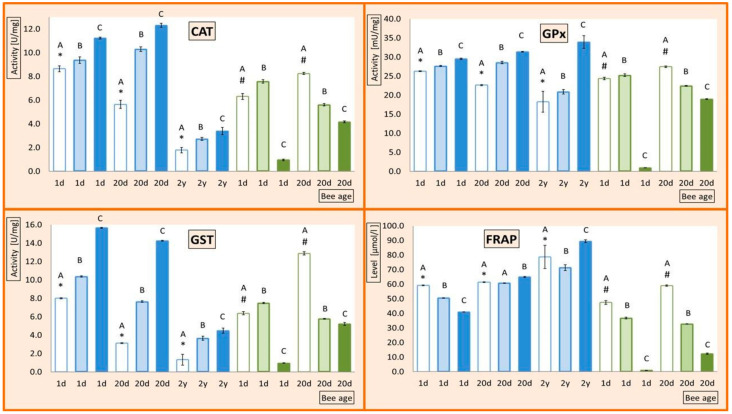
Mean activities (mean ± SD) of catalase (**CAT**), glutathione peroxidase (**GPx**), glutathione S-transferase (**GST**), and the level of the ferric reducing total antioxidant potential (**FRAP**). Error bars indicate SD. 1-day-old bees (1d), 20-day-old bees (20d), and 2-year-old bees (2y); queens (**blue bars**); workers (**green bars**); bee diet contains 0 ppm (**transparent bars**), 5 ppm (light-shadowed bars), and 200 ppm (dark-shadowed bars) imidacloprid. Different capital letters indicate that differences between the means (diets) nested within a particular age and caste are significant (ANOVA + LSD; *p* < 0.01). Values marked with # differ significantly from each other. Values marked with asterisk differ significantly from each other.

**Figure 3 animals-11-01246-f003:**
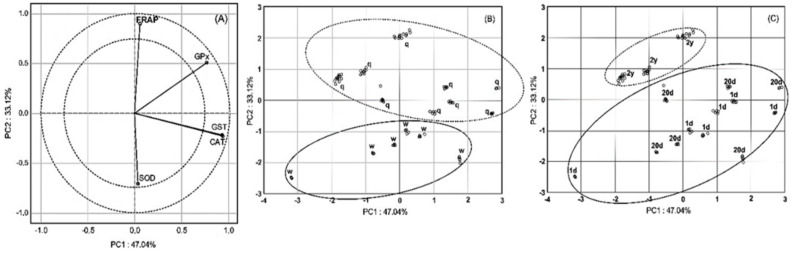
Principal component analysis (PCA): (**A**) two-dimensional projection catalase (CAT), glutathione peroxidase (GPx), glutathione S-transferase (GST), superoxide dismutase (SOD), and ferric reducing antioxidant potential (FRAP) in the principal components PC1 and PC2; (**B**) projection of workers (w; solid) and queens (q; dotted) in the principial components PC1 and PC2; (**C**) two-dimensional projection of the age classes (dotted–2-year-old; queens only, solid–1-day-old and 20-day-old insects in the principal components PC1 and PC2 planes.

## Data Availability

Raw data are available on request from corresponding author.

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
