# Peer review of "Antioxidation Defenses of Apis mellifera Queens and Workers Respond to Imidacloprid in Different Age-Dependent Ways: Old Queens Are Resistant, Foragers Are Not"

_animals, 2021, doi:10.3390/ani11051246_

Round 1

Reviewer 1 Report

The Authors investigated how were different antioxidant defenses (ADSs) shaped by evolution in young/old Apis mellifera workers and queens to broaden the limited knowledge on whether ADSs  are effective in environments contaminated by pesticides. 
The ADS activity was determined in 1-day-old, 20-day-old, and 2-year-old queens and in 1-day-old and 20-day-old workers (foragers) fed without (control) or with low or high imidacloprid (in bee food). ADSs was upregulated in workers with age but downregulated in queens.  Imidacloprid oxidative stress suppressed the active ADSs in workers, particularly 20-day-old foragers, but not in 1-day-old queens.
The Authors have shown that queen and worker ADSs respond to imidacloprid in opposite ways, and old queens were still resistant but 
foragers were not. This may be unfavorable for foragers dwelling in ecosystems that expose them to pesticides. 

I believe that the manuscript should be improved in language, as some sentences are not very understandable. However, the research is original and very interesting and therefore I recommend its publication.

Author Response

Responses to the Reviewer 1; Animals-1186188; 21.04.2021

Manuscript title:  “Antioxidation defense of Apis mellifera queens and workers responds to imidacloprid in different, age-dependent way: old queens are resistant, foragers are not”.

We have not been able to find any other comments in the opinion  sent by Reviewer 1.

Authors: Reviewer 1  suggested minor language correction. Therefore. I sent the ms. to the US for the professional editing at San Francisco Edit; the certificate of which is attached. A few paragraphs I have reedited after they had been edited by a Polish certified editor.

We have not been able to find any other comments in the opinion of Reviewer 1.

Reviewer 2 Report

This study by Paleolog et al. tackles an interesting question: How does pesticide exposure affect the abundance of enzymes involved in oxidative stress response in honey bee queens and workers. This study has 2 interesting aspects: Firstly, it assesses the physiological response to oxidative stress in longlived vs shortlived individuals and secondly, investigates the consequences of pesticide usage (ID in particular) on pollinator physiology in a more “applied sciences viewpoint”. I have some concerns especially regarding how the samples were produced which should be clarified. Furthermore, I have some suggestions for how the results can be interpreted which I hope the authors will find helpful.

Major comments:

  • The introduction requires some revision. In particular, 1) a more comprehensive review of why (evolutionary speaking) extended longevity of reproductive individuals in social insects evolved. Whereas this might be clear for the authors and others working on social insects, but it might not be for scientists not working on eusocial insects. 2) Regarding the Hypothesis 1 and 2 (Line 64-70): The hypothesis is not well introduced and not well explained yet. It is unclear to me what the “thought” difference in ADS functioning is (is there a reference the authors have in mind?), what exactly the authors mean by “ADS functioning” (the underlying molecular mechanisms?) and why is should be different between old workers and old queens? I suspect the authors mean that ADS mechanisms are more active in queens compared to workers. Is that correct? 3) The authors should be more specific about what they mean by “anthropogenically changed environments”. 4) A justification of why these five enzymes have been quantified and whether or not their function is redundant could help to better understand the idea behind the paper.
  • Figure 1-2: In the figure caption, it says that “Different capital letters […] are significant”. However, I can’t find those letters in the figure. Is it possible that the authors missed to add them to the figure or am I missing something?
  • I am somewhat puzzled why the authors used a PCA to analyse the data in figures 1-3. I think a GLM using the enzyme activity as a response variable and caste, ID concentration, age and their interaction as explanatory factors and subsequent posthoc-tests could work better and would be more appropriate. Can the authors please elaborate on why they used PCA?
  • Which tasks were the workers performing at the respective sampling ages? If the workers differ in task choice (e.g. brood carers vs. foragers), the authors did not only compare workers that differ in age but also in task. This is important as task/caste influences gene expression independently from age (e.g. Lucas et al. 2017: Gene expression is more strongly influenced by age than caste in the ant Lasius niger; Kohlmeier et al. 2019: Gene expression is more strongly associated with behavioural specialization than with age or fertility in ant workers).
  • Were those queens samples for these studies virgins? I assume that at least those at day 1 did not mate yet. Mating triggers dramatic changes in female insects and is also involved in activating the ovaries. That what extent might have differences in mating status impacted the findings?
  • I do not agree with the statement that “eusocial evolution produced distinctly different life-span phenotypes for queens and workers”(Line 233). In eusocial insects, unlike in other animal species, reproduction and longevity are positively correlated with each other which can for instance be observed in the fact that ovarian activation of worker results in higher longevity. I thus do not think that queens and workers differ in longevity because they express different “longevity phenotypes” but because they differ in their reproductive activity. This is an aspect that occurs again later on in the discussion.
  • It seems surprising to me that in queens, the concentration of all enzymes except FRAP are decreasing with age. This is against what I would predict given the high longevity of queens and the suggested involvement of oxidative stress response in this. Do the authors have an explanation for this? In line with this, the authors state in line 236, that “one should expect distinct age-dependent differences between the activities of ADSs in workers and queens” and that “our study has confirmed this assumption”. Given the differences in longevity between queens and workers, I would predict that the ADSs are stable in queens but decrease with age in workers. In line with this: Maybe not the baseline abundance of these ADS genes is important for longevity but how strongly they can be upregulated when exposed to oxidative stress? That could explain why in figure 2, most quantities increase upon ID exposure in longlived queens while this effect is absent in workers (at least the foragers)
  • I think one point that could make the authors main conclusion (ID is harmful through a downregulation of ADSs) much stronger are data that show an actual fitness effect of ID exposure and the downregulation of ADSs: Did those workers die earlier? Did they display a decrease in task performance? Did the queen lay less eggs? Where the workers that hatched in ID exposure smaller or less viable? Something like this could link ID exposure and ADS quantification to fitness
  • The finding that most ADS are downregulated upon ID exposure in workers but upregulated in queens is for me the most interesting finding of the study and it deserved are more comprehensive discussion. For instance, how can this result be explained in terms of colony fitness? To what extent can this finding be linked to food provision (The queen received more food than workers). Moreover, the queen consumes more protein than workers which is required for oogenesis. Do the authors think that this additional protein food might have diluted out the ID (which was provided through sucrose solution)? I also find it interesting that in the young workers (which are typically more fertile than old workers), the effect of ID seems to be more queen-like than forager-like. This is in line with RNAseq studies that found that broodcarer gene expression is more similar to queen gene expression than to forager gene expression (See. e.g. Harrison et al. 2015: Reproductive workers show queenlike gene expression in an intermediately eusocial insect, the buff-tailed bumble bee Bombus terrestris) I don’t know whether the stats backup this observation but it would add another interesting point to the paper
  • Regarding lines 287-296: Again (see my points 5, 6, 7 and 9), I think discussing these findings it the light of the respective reproductive activity makes more sense than in the light of different “functionality of ADS”

Minor comments:

To increase the relevance of the study, maybe the authors can give some number of how frequently ID is used? If it’s a globally used chemical, this study is for sure more relevant than if ID is only a niche compound

Line 62: Life-HIstories?

Line 76: “variation” is a better word than “diversity” in this context

Line 86: Im not an expert here, but is the description of ID as an “oxidative stressor” correct here? Has it been shown to induce oxidative stress? A downregulation of antioxidant genes and a shorting of life-span are to me not evidence enough for such a classification

Line 95: What is the justification for using these concentrations? What is the concentration that bees are exposed to in an agricultural environment?

Line 249: What does “function well” mean?

Line 281: The authors should consider including the study by Negroni et al. 2021 (Experimental increase in fecundity causes upregulation of fecundity and body maintenance genes in the fat body of ant queens) here

Author Response

Responses to the Reviewer 2; Animals-1186188; 21.04.2021

Manuscript title:  “Antioxidation defense of Apis mellifera queens and workers responds to imidacloprid in different, age-dependent way: old queens are resistant, foragers are not”.

Authors Reviewer 2 moderate language correction. Therefore. I sent the ms. to the US for the professional editing at San Francisco Edit; the certificate of which is attached. A few paragraphs I have reedited after they had been edited by a Polish certified editor.

Major comments - Reviewer 2:

Authors: To meet the Reviewer suggestions new, proper references have been added. Line  559-666.

R2-1. The introduction requires some revision. Authors: The introduction has been revised. The explanations are given below.  (please see Lines  69-79).

R2-1.1 a more comprehensive review of why (evolutionary speaking) extended longevity of reproductive individuals in social insects evolved. Whereas this might be clear for the authors and others working on social insects, but it might not be for scientists not working on eusocial insects. Authors: We have decided to give this comprehensive review in the Discussion (Lines 357-368)  considering the Reviewer’s remarks, as those issues are considered also in R2-7., R2-8.1., R2-8.2., and R2-8.3.

R2-1.2. Regarding the Hypothesis 1 and 2 (Line 64-70): The hypothesis is not well introduced and not well explained yet. It is unclear to me what the “thought” difference in ADS functioning is (is there a reference the authors have in mind?),

Authors: The paragraph has been reedited (Lines .  We agree; “thought” is a very unclear word here.

We meant that it had been proved in earlier studies that the previous evolution shaped ADSs in queens and workers in  different ways.

But in the corrected version we used the phrase : ” we hypothesize that these ways are not just different but very different and even opposite (contrast). This has been clarified in ms. (see Lines  69-77)      

R2-1.3. what exactly the authors mean by “ADS functioning” (the underlying molecular mechanisms?) and why is should be different between old workers and old queens? I suspect the authors mean that ADS mechanisms are more active in queens compared to workers. Is that correct?

 Authors: That is correct. The Reviver is in 100% right. “functioning” is very imprecise word in this context. This has been corrected and clarified throughout not only in Introduction but in the entire manuscript; phrases like “perform better”; “are more active”; etc. have been used.    

R2-1.4. The authors should be more specific about what they mean by “anthropogenically changed environments”.

Authors: the phrase “anthropogenically changed environments” and the word  “anthropogenic” have been clarified/replaced over the entire manuscript.

For instance the phrase “anthropogenically changed environment” has been replaced by the phrase “anthropogenically contaminated (e.g. by pesticides) environment” that is more clear and fits  the manuscript better.

R2-1.5 A justification of why these five enzymes have been quantified and whether or not their function is redundant could help to better understand the idea behind the paper.

Authors: the following paragraph has been added to M&M in the lines 175-177:

“CAT, GPx, GST, and SOD are the major ROS scavenging and antioxidant enzymes in honey bees [18, 19], and they constitute the first line of their ADS barrier.”

This issue has been developed (the interpretation of our findings, the idea behind the paper) by adding the following paragraph in Discussion: 

“To better support the above findings we should consider the following: CAT, GPx, GST, and SOD are the major ROS scavenging, antioxidant enzymes in honey bees constituting the first line of the ADS barrier against xenobiotic oxidative stress factors. Their functions are limited to only this activity and their secretion/activation is directly con-trolled by ROS. In addition to antioxidation, the other enzymes , e.g. vitellogenin, regulates other physiological processes and are adjusted by hormones, constituting the second ADS line [5,6,2830,42].The first line of defense in our foragers was suppressed by ID, even though its activity was increased by aging. On the contrary, the activity of the first line of defense in 2-year-old queens decreased with age. However, it was not suppressed but rather only "silenced", as its activity retained the ability to be increased in response to ID. It is the age-related downregulation of the ADS enzymes in our queens, accompanied by FRAP upregulation, that is in line with this finding. This is very interesting, new evidence that requires further research. Consequently, it may be not so much the baseline activity of ADS that is important for longevity as how strongly the characteristics can be upregulated when exposed to oxidative stress.”.

R2-2. Figure 1-2: In the figure caption, it says that “Different capital letters […] are significant”. However, I can’t find those letters in the figure. Is it possible that the authors missed to add them to the figure or am I missing something?

R2-3. I am somewhat puzzled why the authors used a PCA to analyse the data in figures 1-3. I think a GLM using the enzyme activity as a response variable and caste, ID concentration, age and their interaction as explanatory factors and subsequent posthoc-tests could work better and would be more appropriate.

Authors:

The devil must have covered my keyboard with his tail when I was copying Figures 1 and 2; I lost the phrase "different capital letters". I am ashamed of this. This may have been the reason why the Reviewer realized that ANOVA (GLM) was not done. But it was  done, of course. Moreover, the section "2.5. Statistical Analysis" first describes the PCA used in Fig. 3 and then the ANOVA used in Figs. 1, 2. This also might caused a misunderstanding. Therefore, the order of the description has been changed (Lines 185-189 were transferred to Lines 179-182) . In the captions of Figs. 1 and 2 the information that ANOVA results were used in their construction has been added. The previous  Figs. 1 and 2 have been replaced by the new corrected Figs. 1 and 2 in the manuscript. Now everything should be clear.

We would like to point out the we have presented the most important evidence from the ANOVA, which might have been overlooked by the Reviewer; “ANOVA confirmed that the caste effects were the greatest (SOD, CAT, GST, GPx, FRAP; F = 59.50, 46.88, 12.55, 21.92, 41.61; df = 1; P = 0.0000, 0.0000, 0.0008, 0.0000, 0.0000). The effects of age were significant, but involved in age × caste interactions. Effects of ID dose were not so clear, and often insignificant, as they were involved in caste × ID dose and age × ID dose interactions. This is in agreement with the PCA results.”   

Our ANOVA delivered dozens of graphs and tables (3 factors, their interactions x 5 separate characteristics). The approach is the following: We use different statistical procedures taking into account as many factors as possible. Then we select the simplest procedure that adequately explains the phenomenon under consideration, which we treat as "the best" procedure. Simple models are easier to re-test the problem, especially to check the evidences in the future. They also make the main message of the paper easier to understand and therefore possess a certain "beauty" that more complex procedures, like ANOVA lack. PCA, which concentrates on a lot of different information (even on their correlations) within only a few graphs, meets these expectations. Therefore we have used ANOVA where it was useful and necessary (FIGS 1,2) but PCA to visualize and prove the main messages of the paper (Fig. 3).

R2-3. Can the authors please elaborate on why they used PCA?

Authors: One of the characteristic features of the main components is the fact that we are able to determine them according to their decreasing level of variation. In turn, eigenvalues are a measure of the level of variability. In addition, in some cases it is possible to assign  a specific physical, chemical or biological interpretation to the main components or groups of objects. Based on the appropriate criteria, it can be assessed what part of the variability in the data is random, related to the errors made in the measurement, and which results from the nature of the data. The PCA method assumes that the total variability of the data contained is the sum of the specific variability resulting from the nature of the objects and the random variability resulting from measurement errors or the noise of experimental techniques. It is generally assumed a priori that the intrinsic variation, taking into account in the first few main components, is numerically much larger than the undesirable variation. Because the eigenvalues of components contain information about the volatility taken into account by a given component, they are thus a source of data when defining the criterion of materiality of components. It should be emphasized, however, that no good enough criterion has been developed whose application will clearly determine the number of significant components. In the presented research, the analysis of variance, simple correlations, and principal component analysis were carried out at the significance level α = 0.05. The analysis of many variables allows us to capture the differences omitted when analyzing two parameters. The main method in this case is the analysis of the principal components (PCA), which consists in treating the data set as a point cloud in the K-dimensional space, where K is the number of variables. The data cloud is subjected to such a change in coordinates to maximize the variance of the first new coordinate; these procedures are carried out for each subsequent coordinate.

R2-4. Which tasks were the workers performing at the respective sampling ages?

Authors: Throughout the manuscript we call 20-day-old workers “foragers” to indicate their task. Therefore, we only have provided the information that the 1-day-old workers (just emerged) do not perform any special task yet.

The additional proper paragraph (Lines 250-254), which clarify this issue has been also added in Discursion (see it in R2-5),

R2-5. If the workers differ in task choice (e.g. brood carers vs. foragers), the authors did not only compare workers that differ in age but also in task. This is important as task/caste influences gene expression independently from age (e.g. Lucas et al. 2017: Gene expression is more strongly influenced by age than caste in the ant Lasius niger; Kohlmeier et al. 2019: Gene expression is more strongly associated with behavioural specialization than with age or fertility in ant workers).

Authors: Honeybee tasks (behavioral specializations) are more closely linked with age than in some ants [see e.g. (sit); …conventional Vg was unlinked to behavioral specialization, age or fertility, which contrasts to studies on bees and some ants; in  Kohlmeier et al. 2019]. Our-20-day old workers were just  starting the forager function. The apian phenotypic plasticity allows workers being at the same age to perform different behavioral tasks but in the such situations the biological senescence is slowing down, accelerated or even reversed [Münch at all 2008: Aging in a eusocial insect: molecular and physiological characteristics of life span plasticity in the honey bee]. Therefore the biological senescence is something different than ageing counted in days. Our conclusions concern rather the biological senescence than ageing expressed in days. On the other hand, analyzing the gene transcripts (intermediate product) – like Lucas et al. 2017 and Kohlmeier et al. 2019 did is useful. However, we studied the enzyme activities (final product) which are essential for survival/fitness. Summing up, we decided to concentrate on the main message of this paper and  not to develop the Discussion, as it could be too speculative here.  However, to meet the Reviewer comments we have added the additional at the beginning of the Discussion (Lines 251-261) which makes our reasoning more concise and precise:

R2-6. Were those queens samples for these studies virgins? I assume that at least those at day 1 did not mate yet. Mating triggers dramatic changes in female insects and is also involved in activating the ovaries. That what extent might have differences in mating status impacted the findings?

Authors: the necessary information has been added in M&M and in the Discussion (Lines 254-257). See again R2-5.   

R2-7. I do not agree with the statement that “eusocial evolution produced distinctly different life -span phenotypes for queens and workers”(Line 233).

Authors: The “life-span phenotypes” or “longevity phenotypes” are terms used when queens and workers are compared [e.g. Münch at all 2008: Aging in a eusocial insect: molecular and physiological characteristics of life span plasticity in the honey bee].Phenotype is the observable characteristics or traits of an organism that are produced by the interaction of the genotype and the environment. Thus longevity, as the trait value, is the phenotype, and it is certainly distinctly different in workers and queens. For millions years of evolution [e.g. references 29 and 2, line 234, in our ms.] we must consider the co-evolution of longevity, fecundity, and sociality in insects. Overlap of generations was pivotal there. It occurs when the preliminary differences in the female longevity exists [see; Johnson & Carey (2014): Hierarchy and connectedness as determinants of health and longevity in social insects]. So different life-span phenotypes were necessary as the initial trigger factor. A female (future queen) as it was much more long lived than the helper-females (future workers) started to monopolize oviposition; shorter living helper-females had emerged, which had a smaller contribution to reproduction. The colony increase deepened this specialization; emergence of the super-organism. In fact we face here the complicated mechanism of mutual co-evolution of fecundity and longevity that involve compromises and  feedbacks. Therefore, when we think of the life-span phenotype we think also about the reproductive phenotype.

The reviewer is right; the current research shows that the fecundity was basic, but it was positively corelated with longevity. One way or another, this is the eusocial evolution that produced distinctly different life-span phenotypes for queens and workers. The questions concern only the trigger factors, the traits directly exposed to the natural selection and the genetic or functional correlations between them. It does not matter whether a given trait was selected directly or by the functional or genetic correlation with the others (correlated response to selection).

We have developed this issue a little bit in the Lines 279-281 and also in the separate paragraph at the end of the Discussion (see Minor comments – the last paragraph; (Lines 358 394).        

R2-8.1 In eusocial insects, unlike in other animal species, reproduction and longevity are positively correlated with each other which can for instance be observed in the fact that ovarian activation of worker results in higher longevity. I thus do not think that queens and workers differ in longevity because they express different “longevity phenotypes” but because they differ in their reproductive activity. This is an aspect that occurs again later on in the discussion.

Authors: Maybe there is the problem in how we understand term “phenotype” in a different way. Again: “phenotype” is the observable characteristics or traits of any organism that are produced by the interaction of the genotype and the environment (broadly understood). Therefore, , when the workers and queens differ in longevity, they at the same time express different “longevity phenotypes”. We can also consider different “ADS phenotypes”. Despite identical genomes, bees of each caste have very different life spans (i.e. the longevity phenotypes). These differences are linked to the reproductive abilities and thought to be at least partially attributable to their varied ability to cope with oxidative stress (ADS phenotypes). Of course, reproductive ability (fecundity phenotype) is important (see comments to R2-7) but we would like to focus in this discussion on the link between longevity phenotype and ADS phenotype. The longevity phenotype, ADS phenotype, and other phenotypes constitute the total phenotype, e.g. of a queen (population genetics). For more explanations see Minor comments – the last paragraph. The relevant information are added to ms (Lines 279-287; 357-366)

R2-8.2. It seems surprising to me that in queens, the concentration of all enzymes except FRAP are decreasing with age. This is against what I would predict given the high longevity of queens and the suggested involvement of oxidative stress response in this. Do the authors have an explanation for this?

Authors: We  have also found this finding as a little bit strange and surprising, and therefore, new and interesting. However, we tried to explain it in the Discussion. Has it been overlooked? Following the Reviewer’s suggestion we re-edited this paragraph to make it more clear and convincing. Now it sounds as follow:

“Another important finding here is that the downregulation of the ADS enzymes with aging in our queens was accompanied by FRAP upregulation”………… “the ADSs of our queens have to work differently and may involve many more different components than the ADSs of our workers during exposure to ID stress” …….. and, finally, the entire paragraph from R2-1.5: “To better support the above findings we should to consider the following: CAT, GPx, GST, and SOD are the major ROS scavenging, antioxidant enzymes in honey bees being the first line of the ADS barrier against xenobiotic oxidative stresses…….. and so on”……. Followed by the sentence:  “Consequently, maybe not the baseline activity of ADS is important for longevity but how strongly they can be upregulated when exposed to oxidative stress?”

R2-8.3. I do not agree with the statement that “eusocial evolution produced distinctly different life In line with this, the authors state in line 236, that “one should expect distinct age-dependent differences between the activities of ADSs in workers and queens” and that “our study has confirmed this assumption”. Given the differences in longevity between queens and workers, I would predict that the ADSs are stable in queens but decrease with age in workers. In line with this: Maybe not the baseline abundance of these ADS genes is important for longevity but how strongly they can be upregulated when exposed to oxidative stress?

That could explain why in figure 2, most quantities increase upon ID exposure in longlived queens while this effect is absent in workers (at least the foragers).

Authors: The Reviewer may be is right, but to the best of our knowledge this is the first such  finding. Therefore, we would like to avoid overinterpretation. However, encouraged by the Reviewer we have introduced to the Discussion (see R2-8.) his following assumption: “Maybe not the baseline activity of ADS is important for longevity but how strongly they can be upregulated when exposed to oxidative stress?” accompanied with the proper comment. The entire paragraph (Line 357-391) concerns this issue – among others.

R2-9. I think one point that could make the authors main conclusion (ID is harmful through a downregulation of ADSs) much stronger are data that show an actual fitness effect of ID exposure and the downregulation of ADSs: Did those workers die earlier? Did they display a decrease in task performance? Did the queen lay less eggs? Where the workers that hatched in ID exposure smaller or less viable? Something like this could link ID exposure and ADS quantification to fitness.

Authors: In the Introduction we have written that: “ID, which honey bees are exposed to by agriculture, has been found to be harmful to honey bees’ health [11–13], including their detoxification abilities and reproductive functions [14–17]. Moreover, ID impairs the honey bee ADS and suppresses the antioxidant genes [3], simultaneously shortening workers’ life-spans [18,19].” Therefore, in the Introduction we have presented the evidence documenting that ID decrease the colony fitness . Has that been overlooked or not clear enough? Hence, to better emphasize this issue, the next paragraph of the Introduction has been re-edited as follows: “ID, which honey bees are exposed to by agriculture, has been found to be harmful to honey bees’ field navigation and health [11–13], including their detoxification abilities and reproductive functions [14–17]. Moreover, ID impairs the honey bee ADS and suppresses antioxidant genes [3],………. and shortening workers’ life-spans [18,19]. Consequently, ID reduce the colony fitness.”   

In the Discussion the following statements have been presented as well:

  • Therefore, the cytoplasmic and SOD-dependent mitochondrial conversion of O2 to H2O2 and O2 may be impaired by ID independent of bee caste and age, which is important for the fitness of the entire colony.
  • Thus, decreased forager ADSs due to decreasing colony survival may impact the colony fitness (compare) Lemanski et al. [10]).
  • Due to impairment of the ADS by ID, longevity [3,11], resistance to parasitoses [34,13], and harmful xenobiotics [35] may also be decreased in the foragers. Thus, decreased forager ADSs due to decreasing colony survival may impact the colony fitness (compare Lemanski et al. [10]).

Summing up, we think that we have confirmed the adverse  impact of ID for the colony fitness both in the Introduction and the Discission.    

The finding that most ADS are downregulated upon ID exposure in workers but upregulated in queens is for me the most interesting finding of the study and it deserved are more comprehensive discussion. For instance,

Authors: The proper paragraph  has been added (Line 357-391) concerns this issue – among others

how can this result be explained in terms of colony fitness?

Authors: The proper paragraph  has been added (Line 357-391) concerns this issue – among others

 To what extent can this finding be linked to food provision (The queen received more food than workers).

Authors: The proper paragraph  has been added (Line 357-391) concerns this issue – among others

Moreover, the queen consumes more protein than workers which is required for oogenesis. Do the authors think that this additional protein food might have diluted out the ID (which was provided through sucrose solution)?

Authors: Therefore, our bees were fed with the pollen candy (proteins) contaminated with ID. So this factor was not important too much.

I also find it interesting that in the young workers (which are typically more fertile than old workers), the effect of ID seems to be more queen-like than forager-like. This is in line with RNAseq studies that found that broodcarer gene expression is more similar to queen gene expression than to forager gene expression (See. e.g. Harrison et al. 2015: Reproductive workers show queenlike gene expression in an intermediately eusocial insect, the buff-tailed bumble bee Bombus terrestris) I don’t know whether the stats backup this observation but it would add another interesting point to the paper.

Authors: The proper paragraph  has been added (Line 357-391) concerns this issue – among others

Consequently, one more conclusion has been added (Line 412-413).

R2-11. Regarding lines 287-296: Again (see my points 5, 6, 7 and 9), I think discussing these findings it the light of the respective reproductive activity makes more sense than in the light of different “functionality of ADS”

Authors: The proper paragraph  has been added (Line 357-391) concerns this issue – among others

Minor comments:

To increase the relevance of the study, maybe the authors can give some number of how frequently ID is used? If it’s a globally used chemical, this study is for sure more relevant than if ID is only a niche compound

Authors: The paragraph has been developed as originally it had included only in the reference [12] .

The previous  version was:Imidacloprid (ID) [1-(6-chloro-3-pyridylmethyl)-2nitroimino-imidazolidine], which honey bees are exposed to by agriculture, has been found to be harmful to honey bees’ health [11–13], including their detoxification abilities and reproductive functions [14–17]. However, ID impairs the honey bee ADS and suppresses the antioxidant genes [3], simultaneously shortening workers’ life-spans [18,19]. Therefore, we used ID as a common, contemporary oxidative stressor [12].”

The new, developed version is (Lines 91-102): “Imidacloprid (ID) [1-(6-chloro-3-pyridylmethyl)-2nitroimino-imidazolidine], which honey bees are exposed to by agriculture, has been found to be harmful to honey bees’ field navigation and health [11–13], including their detoxification abilities and reproductive functions [14–17]. Moreover, ID impairs the honey bee ADS and suppresses antioxidant genes [3], simultaneously increasing MDA (a marker of oxidative stress; lipid peroxidation) and shortening workers’ life-spans [18,19]. Consequently, ID reduces the colony fitness. The globally used ID, which stimulates numerous discussions and researches, belongs to neonicotinoid insecticides that are implicated as one of the major reasons for declines in global bee populations. Honeybees (essential pollinators worldwide) are susceptible to even potentially sublethal exposure to neonicotinoids [12]. Therefore, we used ID as a common, contemporary oxidative stressor.”

Line 62: Life-Histories?

Authors: this phrase was simplified too much so it has been corrected; The original phrase was:  Honey bee queens and workers have different life-stories, evolving very different life-span phenotypes……..”.

The new, more clear phrase is (Lines 66-68): Honey bee queens and workers have different life-history strategies, evolving very different life-span phenotypes”. 30. [this terminology is in line with those used by Pamplona et al (2011 ) Molecular and structural antioxidant defenses against oxidative stress in animals]

Line 76: “variation” is a better word than “diversity” in this context

Authors: Agree; this has been changed

Line 86: I’m not an expert here, but is the description of ID as an “oxidative stressor” correct here? Has it been shown to induce oxidative stress? A downregulation of antioxidant genes and a shorting of life-span are to me not evidence enough for such a classification

Authors: imidacloprid affects energy production by bees’ mitochondria. Consequently, ROS and the polyunsaturated fatty acids of lipid membranes induce  lipid peroxidation (LPO). The end product of these reactions is malondialdehyde (MDA), a marker of oxidative stress. Imidacloprid significantly increases MDA, and therefore causes the oxidative stress; not only in bees.

The proper, additional remark accompanied with the reference [Balieira et al. (2018) Imidacloprid-induced oxidative stress in honey bees and the antioxidant action of caffeine] has been added in ms. The final version is (Lines 94-102):Moreover, ID impairs the honey bee ADS and suppresses antioxidant genes [3], simultaneously increasing MDA (a marker of oxidative stress; lipid peroxidation) and shortening workers’ life-spans [18,19]. Consequently, ID reduces the colony fitness. The globally used ID, which stimulates numerous discussions and researches, belongs to neonicotinoid insecticides that are implicated as one of the major reasons for declines in global bee populations. Honeybees (essential pollinators worldwide) are susceptible to even potentially sublethal exposure to neonicotinoids [12]. Therefore, we used ID as a common, contemporary oxidative stressor.”

Line 95: What is the justification for using these concentrations? What is the concentration that bees are exposed to in an agricultural environment?

Authors: The following information accompanied by references has been added: “5 ppb is close to field- relevant sublethal residual concentration) whereas 200 ppb is considered potentially lethal [11, 17].” (Line 112-115)

Line 249: What does “function well” mean?

Authors :agree; “function well” sounds unclear. We have made the correction. 

The original sentence was: “To the best of our knowledge, we analyzed the ADS of queens of such an advanced age for the first time and their ABS still functioned well when exposed to ID despite the queen senescence”.

The new, corrected sentence is:To the best of our knowledge, we analyzed the ADS of queens of such an advanced age for the first time and their ABS was still  very effective when exposed to ID despite the queens’ senescence.”

Line 281: The authors should consider including the study by Negroni et al. 2021 (Experimental increase in fecundity causes upregulation of fecundity and body maintenance genes in the fat body of ant queens) here

Authors: the following paragraph has been added to meet the Reviewer’s comment in line 281 and many other in R2-7., R2-8.1., R2-8.2., and R2-8.3., R2-10., and  R2-10.:

The new paragraph has been introduced to the discussion: “The negative link between reproduction and longevity evolved in most solitary animals, as the organism (soma) becomes redundant when offspring has been produced, particularly while there is a shortage of resources that needs a number of trade-offs be-tween body maintenance and reproductive functions. However, this link has been evolutionary decoupled in social insects [2, 31, 50] …… This shows that reproductive abilities, no matter how real or expected, were crucial for the evolutionary shaping of the network of linkages: fecundity-longevity, body maintenance/defense and particular phenotypes of these traits”.

The entire paragraph is placed; Lines 357-391 of the ms.